# SafeMap: Robust HD Map Construction from Incomplete Observations

Xiaoshuai Hao[1]    Lingdong Kong[2]    Rong Yin[3]    Pengwei Wang[1]    Jing Zhang[4]
Yunfeng Diao[5 6]    Shu Zhao[7]

## Abstract

Robust high-definition (HD) map construction is vital for autonomous driving, yet existing methods often struggle with incomplete multi-view camera data. This paper presents **SafeMap**, a novel framework specifically designed to ensure accuracy even when certain camera views are missing. SafeMap integrates two key components: the Gaussian-based Perspective View Reconstruction (G-PVR) module and the Distillation-based Bird's-Eye-View (BEV) Correction (D-BEVC) module. G-PVR leverages prior knowledge of view importance to dynamically prioritize the most informative regions based on the relationships among available camera views. Furthermore, D-BEVC utilizes panoramic BEV features to correct the BEV representations derived from incomplete observations. Together, these components facilitate comprehensive data reconstruction and robust HD map generation. SafeMap is easy to implement and integrates seamlessly into existing systems, offering a plug-and-play solution for enhanced robustness. Experimental results demonstrate that SafeMap significantly outperforms previous methods in both complete and incomplete scenarios, highlighting its superior performance and resilience.

## 1. Introduction

Online high-definition (HD) map construction is a critical and challenging task in autonomous driving, providing pre-

[1]Beijing Academy of Artificial Intelligence [2]National University of Singapore [3]Institute of Information Engineering, Chinese Academy of Sciences [4]School of Computer Science, Wuhan University [5]School of Computer Science, Hefei University of Technology [6]Intelligent Interconnected Systems Laboratory of Anhui Province (Hefei University of Technology) [7]Independent Researcher. Correspondence to: Yunfeng Diao <diaoyunfeng@hfut.edu.cn>.

*Proceedings of the 42nd International Conference on Machine Learning*, Vancouver, Canada. PMLR 267, 2025. Copyright 2025 by the author(s).

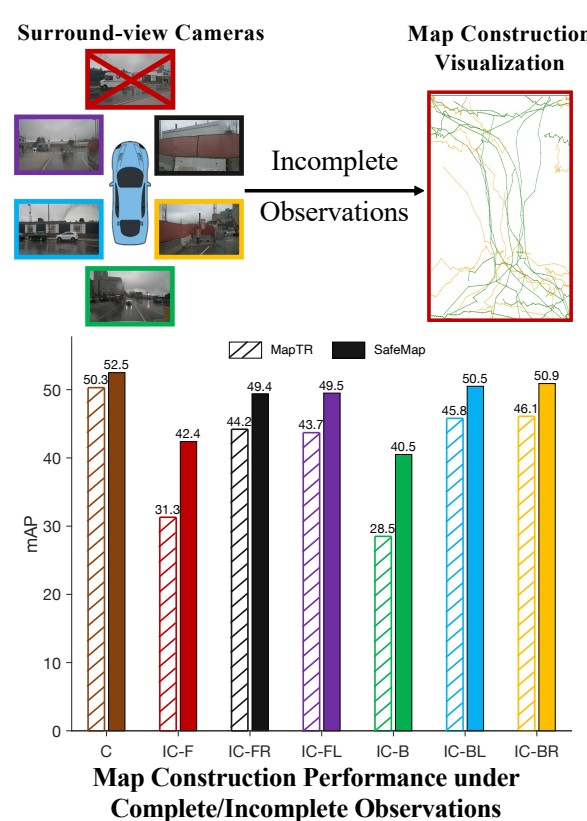

*Figure 1.* **Online Vectorized HD Map Construction with Incomplete Observations.** C/IC denotes complete/incomplete observations. IC-F/FL/FR/B/BL/BR represents a collection of six scenarios of missing camera views, where F, B, L, and R correspond to front, back, left, and right camera views, respectively.

cise and detailed static environmental information essential for vehicle planning and navigation (Caesar et al., 2020; Zhang et al., 2025). These maps enable ego-vehicles to accurately localize themselves on the road and anticipate upcoming features (Zhang et al., 2023; Ding et al., 2023; Qiao et al., 2023). HD maps encapsulate vital semantics, including road boundaries, lane dividers, and road markings (Liao et al., 2023a; Hao et al., 2024a). Depending on the input sensor modality, HD map construction models are typically categorized into three types: camera-based (Hao et al., 2025b; Yuan et al., 2024), LiDAR-based (Li et al., 2022a; Sauerbeck et al., 2023), and camera-LiDAR fusion models (Liao et al., 2023a; Hao et al., 2024b; 2025a).

Recently, multi-view camera-based HD map construction methods have gained attention due to advancements in Bird's-Eye-View (BEV) perception (Chen et al., 2024a; Ma et al., 2024). Compared to LiDAR-based and fusion-based approaches, multi-view camera methods are easier and more cost-effective to deploy (Li et al., 2022b; Liu et al., 2023c). However, a key limitation is their dependence on complete multi-view images, which can lead to catastrophic failures when views are obstructed by occlusions or sensor malfunctions (Kong et al., 2023b; Li et al., 2024). As shown in Fig. 1, the absence of crucial visual information can significantly degrade overall map construction. Thus, it becomes crucial to enhance the robustness of online vectorized HD map construction under incomplete visual observations. Overcoming these challenges is essential for ensuring safe navigation, particularly in complex and extreme driving scenarios, thereby significantly contributing to the overall reliability of autonomous systems (Kong et al., 2024).

Enhancing the robustness of driving perception models against sensor failures has emerged as a prominent topic in autonomous driving research (Kong et al., 2023b;a; Liu et al., 2023a). MapBench (Hao et al., 2024a; 2025c) evaluates the reliability of HD map models against various real-world corruptions, revealing that sensor failures can severely compromise performance and traffic safety. Although recent methods like MetaBEV (Ge et al., 2023) and UniBEV (Wang et al., 2024) have started to tackle camera failures in 3D object detection, maintaining commendable performance even when camera views or LiDAR signals are lost. However, these approaches still rely on complete multi-view camera images and do not tackle the challenges of incomplete observations due to camera damage or occlusions (Xie et al., 2023). Additionally, Chen (Chen et al., 2024b) introduced a masked view reconstruction module (M-BEV) to enhance robustness in 3D object detection under various missing view scenarios. Nonetheless, incomplete multi-view camera-based methods for HD map construction remain underexplored.

To bridge this gap, we propose a novel framework, **SafeMap**, designed to maintain accuracy even when camera views are missing. SafeMap comprises two key components: the Gaussian-based Global View Reconstruction (G-GVR) module and the Distillation-based BEV Correction (D-BEVC) module. G-GVR utilizes Gaussian-based sparse sampling to generate strategic reference points based on view importance, serving as spatial priors for lost views. These reference points facilitate deformable attention, allowing the model to dynamically focus on the most informative regions across available views and efficiently reconstruct missing views. Meanwhile, D-BEVC leverages complete panoramic BEV features to further correct the BEV representations obtained from incomplete observations. Importantly, these components are simple yet effec-

tive plug-and-play techniques that integrate seamlessly with existing pipelines. Experimental results demonstrate that SafeMap significantly outperforms previous methods across both complete and incomplete observations, showcasing superior performance and robustness.

The contributions of this paper are mainly three-fold:

- We present SafeMap, a robust HD map construction framework that ensures high accuracy and reliability even in the presence of missing camera views.

- We introduce two innovative techniques in SafeMap: 1) the Gaussian-based Perspective View Reconstruction module, which utilizes relationships among available camera views to infer missing information through Gaussian-based reference point sampling, and 2) the Distillation-based BEV Correction module to further correct the BEV feature extracted from incomplete observations.

- SafeMap outperforms state-of-the-art methods in both complete and incomplete scenarios, demonstrating superior performance and robustness, thereby establishing a strong baseline for HD map construction research.

## 2. Related Work

**HD Map Construction.** The construction of HD maps is a pivotal area of research in autonomous driving, with existing methodologies categorized by their input sensor modality into camera-based (Liu et al., 2024a; Li, 2024; Chen et al., 2025; Zhao et al., 2025; Zhang et al., 2024), LiDAR-based (Li et al., 2022a; Liu et al., 2023b), and camera-LiDAR fusion (Dong et al., 2024; Liao et al., 2023a;b; Xiaoshuai et al., 2025; Zhou et al., 2024) models. Among these, multi-view camera-based approaches are particularly favored for their ease of deployment and cost-effectiveness (Ma et al., 2024; Chen et al., 2024a). However, these methods often rely on the availability of complete multi-view images, making them susceptible to failures when one or more camera views are compromised, potentially leading to traffic safety incidents (Kong et al., 2024). This limitation highlights the urgent need for robust solutions that can facilitate accurate online HD map construction, even in scenarios where camera views are incomplete. Our research pioneers efforts to enhance robustness in multi-view camera-based HD map construction under conditions of view corruption or failure encountered in real-world environments.

**Robustness against Sensor Failures.** Sensor failures significantly challenge the accuracy of perception systems, posing direct risks to the safety of autonomous vehicles. Consequently, developing robustness against these failures has become a crucial research focus (Dong et al., 2023;

*Figure 2.* **Overview of the SafeMap Framework.** We first extract features from complete multi-view camera images and efficiently transform them into a unified BEV space using view transformations. To simulate emergency scenarios involving camera failures, we employ a Random View Masking (RVM) and recovery scheme. Specifically, we introduce a novel Gaussian-based Perspective View Reconstruction (G-PVR) module and a Distillation-based Bird's-Eye-View Correction (D-BEVC) module to reconstruct the missing view information. Finally, the reconstructed BEV features are processed by a map decoder and prediction heads for HD map construction.

Zhu et al., 2023; Kong et al., 2024; Song et al., 2024; Xie et al., 2025). The resilience of BEV perception has been extensively studied across various applications, including 3D object detection (Song et al., 2025; Liu et al., 2023c; Yin et al., 2024), semantic segmentation (Zhang et al., 2022; Zhou & Krähenbühl, 2022; Xu et al., 2024; Hong et al., 2024; Liu et al., 2024b; Kong et al., 2025; Xu et al., 2025), depth estimation (Kong et al., 2023c; Wei et al., 2023a; Ke et al., 2024), and semantic occupancy prediction (Wei et al., 2023b; Tang et al., 2024). Notably, MapBench (Hao et al., 2024a) evaluates the reliability of HD map models under natural sensor corruptions, highlighting significant performance degradation linked to sensor failures.

**Comparisons with Recent Works.** Recent advancements, such as MetaBEV (Ge et al., 2023), UniBEV (Wang et al., 2024), and M-BEV (Chen et al., 2024b), have begun to tackle sensor failures in 3D object detection frameworks. For instance, MetaBEV (Ge et al., 2023) incorporates a meta-BEV query and an evolving decoder to mitigate the impact of sensor failures, while UniBEV (Wang et al., 2024) is designed to improve robustness against missing modalities. Nevertheless, these approaches still rely on complete multi-view camera images and do not address challenges from incomplete observations due to camera damage or occlusions. Furthermore, M-BEV (Chen et al., 2024b) aims to reconstruct image features from neighboring views when specific camera sensors fail. However, HD map construction heavily depends on static environmental data captured by surrounding cameras, necessitating specialized methods to handle incomplete observations. To our knowledge, this work is the **first** to propose a novel approach for HD map construction that addresses the challenges of incomplete multi-view camera data.

## 3. Methodology

SafeMap presents a robust HD map framework that aims to maintain accuracy even in the presence of missing camera views. During training, we randomly mask and reconstruct

camera views, enabling the reconstruction module within the map encoder to predict features for the missing views during testing. The model architecture, as shown in Fig. 2, consists of four key components: the Map Encoder (Section 3.2), the Gaussian-based Perspective View Reconstruction (G-PVR) module (Section 3.3), the Distillation-based BEV Correction (D-BEVC) module (Section 3.4), and the Map Decoder (Section 3.5).

### 3.1. Preliminaries

To ensure clarity in notation, we first define the symbols and concepts used throughout this paper. Our objective is to develop a novel framework for safe and robust HD map construction capable of processing both complete and incomplete multi-view camera images. This framework predicts vectorized map elements in BEV space, specifically targeting three types of map elements: road boundaries, lane dividers, and pedestrian crossings. Formally, we denote a set of multi-view RGB camera images captured in perspective view as $I = \{I_1, I_2, \ldots, I_N\}$, where $N$ represents the number of camera views. Each image $I_i$ is characterized as $I_i \in \mathbb{R}^{H^{\mathrm{cam}} \times W^{\mathrm{cam}} \times 3}$, with $H^{\mathrm{cam}}$ and $W^{\mathrm{cam}}$ denoting the image height and width, respectively.

### 3.2. Map Encoder

We build our Map Encoder based on the popular HD map construction method MapTR (Liao et al., 2023a), which consists of a 2D feature extractor and a Perspective-View (PV) to BEV transformation module. Specifically, we first use 2D feature extractor (He et al., 2016; Liu et al., 2021) to extract multi-scale 2D features from each perspective view $I = \{I_1, I_2, .., I_N\}$, and outputs complete multi-view PV features $F_{\mathrm{PV}}^{\mathrm{com}} = \{F_{I_1}, F_{I_2}, ..., F_{I_N}\}$. Then, we adopt a 2D-to-BEV feature transformation module (Chen et al., 2022) to map the multi-view PV features $F_{\mathrm{PV}}^{\mathrm{com}}$ into BEV features $F_{\mathrm{BEV}}^{\mathrm{com}}$. The BEV features can be denoted as $F_{\mathrm{BEV}}^{\mathrm{com}} \in R^{H \times W \times C}$, where $H, W, C$ refer to the spatial height, spatial width, and the number of channels of feature

maps, respectively. To mimic the emergency situation of camera failure in the testing phase, we adopt a view masking and recovering scheme in the training phase. Specifically, we randomly mask a certain 2D image feature, name the incomplete PV feature as $F_{\text{PV}}^{\text{incom}} = \{F_{I_1}, F_{I_{\text{mask}}}, ..., F_{I_N}\}$, and then initialize the $F_{\text{PV}}^{\text{incom}}$ using the spatial feature cues around it.

### 3.3. Gaussian-based PV Reconstruction Module

The challenge of reconstructing a missing view from multiple perspectives is critical for ensuring comprehensive environmental perception. Existing approaches rely on cropping partial pixels from the left and right adjacent views and processing them through a transformer-based model for missing view reconstruction. While effective, it is constrained by the need for a predefined crop ratio and fails to fully leverage information from all available views, potentially overlooking valuable contextual cues.

To address the aforementioned limitations, we introduce a novel Gaussian-based Perspective View Reconstruction (G-PVR) module, consisting of a learnable query vector that serves as a flexible representation of the missing view coupled with multi-view values derived from all available perspectives. Specifically, G-PVR samples the feature at locations of reference point $p$ to construct keys and values for multi-head attention modules:

$$\hat{k} = \hat{x}W_k, \quad \hat{v} = \hat{x}W_v,$$
$$\text{with } \Delta p = \theta_{\text{offset}}(V), \hat{x} = \phi(F_{PV}^a; p + \Delta p), \quad (1)$$

where $V$ is a learnable query. $W_k$ and $W_v$ are transformation matrices. $\hat{k}$ and $\hat{v}$ are the key and value embeddings. $\theta_{\text{offset}}$ is a light weight network to generate offsets. $\phi$ is a sampling function from deformable attention (Zhu et al., 2021). $F_{\text{PV}}^a = F_{\text{PV}}^{\text{incom}} \setminus \{F_{I_{\text{mask}}}\}$ is the feature of all available views. A naive solution is to generate vanilla reference points from a uniform grid, which, however, ignores the important fact that different views contribute unequally to the reconstruction of the missing view. Adjacent views typically contain the most relevant information and should be given higher importance, while more distant views (*e.g.*, the opposite of the missing perspective) may contain less information and should be weighted accordingly.

Motivated by this, the G-PVR module incorporates prior knowledge about view importance to dynamically focus on the most informative regions, as illustrated in Fig. 3. Specifically, using the left and right views of the missing perspective as a starting point, we tiled all available frames, based on the spatial distance to the missing frame, to construct a panoramic perspective view as $F_{\text{PPV}} = \text{Concat}(F_{\text{PV}}^a) \in R^{H \times N_a * W \times C}$, where $N_a$ denotes the number of available views and concat is the concatenation operation. Then, a set of Gaussian-based reference points is generated to cover

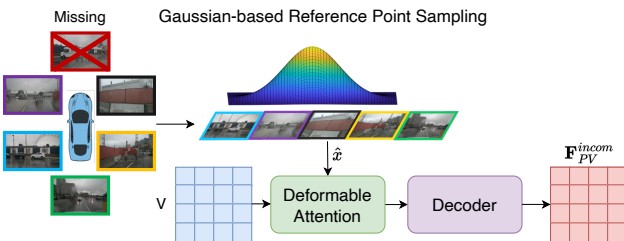

*Figure 3.* Illustration of the proposed Gaussian-based Perspective View Reconstruction (G-PVR) module.

the panoramic perspective view:

$$p_x \sim \mathcal{N}(N_a * W/2; \sigma^2),$$
$$p_y \sim \mathcal{U}(0, H), \quad (2)$$

where $\mathcal{N}$ and $\mathcal{U}$ is the Gaussian distribution and uniform distribution, respectively. $\sigma^2$ is the variance.

G-PVR module effectively guides the module in focusing on the most informative regions of the input views, and can easily adapt to different numbers of input views and varying spatial relationships between views. Through the deformable attention module with Gaussian-based reference points and the panoramic perspective view, the learnable query $V$ aggregates the information from available views according to Eq. (1). Then, we use MAE-like transformer blocks to reconstruct the missing view:

$$\mathcal{L}_{\text{Rec}} = \| F_{\text{PV}}^{\text{com}} - F_{\text{PV}}^{\text{incom}} \|, \quad (3)$$

where $F_{\text{PV}}^{\text{incom}} = \text{Decoder}([V, F_{\text{PPV}}])$. Then, we use a PV-to-BEV feature transformation module to map the incomplete PV feature as $F_{\text{PV}}^{\text{incom}}$ into the BEV feature $F_{\text{BEV}}^{\text{incom}}$.

### 3.4. Distill-based BEV Correction Module

In addition to reconstructing the perspective features of partially missing views, we also need to establish a global BEV feature learning space to further correct the extraction of BEV features from incomplete observations. Specifically, we leverage the complete panoramic BEV features $F_{\text{BEV}}^{\text{com}}$ as the supervisory signal to correct the BEV features of incomplete observations $F_{\text{BEV}}^{\text{incom}}$ via an MSE loss:

$$\mathcal{L}_{Cor} = \text{MSE}(F_{\text{BEV}}^{\text{com}}, F_{\text{BEV}}^{\text{incom}}). \quad (4)$$

We use $\mathcal{L}_{\text{Cor}}$ as one of the correction objectives to enable the panoramic BEV features of incomplete observations can implicitly benefit from the panoramic BEV features of complete observations during the training phase.

### 3.5. Map Decoder

We follow (Liao et al., 2023a) and adopt its BEV feature decoder, composed of several decoding layers and a prediction head. Each decoding layer learns through self-attention

and cross-attention. Self-attention is utilized to facilitate interaction between hierarchical map queries in a decoupled manner. Cross-attention in the decoder is specifically designed to enable interaction between map queries and input BEV features. Specifically, the input to the BEV Decoder is the panoramic BEV features from incomplete observations $F_{BEV}^{incom}$. Then, the Map head (MapHead) employs the classification and point branches to produce the final predictions of map element categories and point positions.

### 3.6. Overall Optimization Objective

To improve the accuracy of online construction of vectorized HD maps in the presence of incomplete observations, we integrate the map loss $\mathcal{L}_{\mathrm{map}}$ with the above reconstruction losses, including the Perspective View Reconstruction loss ($\mathcal{L}_{\mathrm{Rec}}$) and the BEV Feature Correction loss ($\mathcal{L}_{\mathrm{Cor}}$). The overall training objective can be formulated as:

$$L = L_{\mathrm{map}} + \lambda_1 L_{\mathrm{Rec}} + \lambda_2 L_{\mathrm{Cor}}, \quad (5)$$

where $\lambda_1$ and $\lambda_2$ are hyper-parameters for balancing these losses. $\mathcal{L}_{\mathrm{map}}$ is calculated following (Liao et al., 2023a), which is composed of three parts, *i.e.*, classification loss (Mukhoti et al., 2020), point2point loss (Malkauthekar, 2013), and edge direction loss (Liao et al., 2023a).

## 4. Experiments

### 4.1. Experimental Settings

**Datasets.** The nuScenes dataset (Caesar et al., 2020) contains $1,000$ sequences of recordings collected by autonomous driving cars. Each sample is annotated at 2Hz and includes six camera images covering the $360°$ horizontal FOV of the ego-vehicle. For fair evaluation, we follow (Liao et al., 2023a; Zhou et al., 2024) and focus on three types of map elements: pedestrian crossings, lane dividers, and road boundaries. The Argoverse2 dataset (Wilson et al., 2021) consists of $1,000$ logs, each capturing 15 seconds of 20Hz RGB images from 7 cameras, 10Hz LiDAR sweeps, and a 3D vectorized map. The dataset is split into 700 logs for training, 150 logs for validation, and 150 logs for testing. Consistent with previous works (Liao et al., 2023a), we report results on the validation set and focus on the same three map categories as the nuScenes dataset.

**Evaluation Metrics.** We use metrics consistent with previous HD map works (Liao et al., 2023a; Li et al., 2022a; Zhou et al., 2024). Average precision (AP) is used to assess map construction quality, while Chamfer distance ($D_{\mathrm{Chamfer}}$) measures the alignment between predictions and ground truth. We calculate $AP_\tau$ under various $D_{\mathrm{Chamfer}}$ thresholds ($\tau \in T = 0.5\mathrm{m}, 1.0\mathrm{m}, 1.5\mathrm{m}$), averaging across all thresholds to obtain the final mean AP (*mAP*) metric. The perception ranges are $[-15.0\mathrm{m}, 15.0\mathrm{m}]$ and $[-30.0\mathrm{m}, 30.0\mathrm{m}]$ for

*Table 1.* Performance comparisons with (Liao et al., 2023a) when losing each of six camera views on the nuScenes validation set.

| Standard | Method | $AP_{ped.}$ | $AP_{div.}$ | $AP_{bou.}$ | mAP |
|---|---|---|---|---|---|
| All Views | MapTR | 46.3 | 51.5 | 53.1 | 50.3 |
|  | **Ours** | **48.1** | **54.3** | **55.3** | **52.5** |
| **View Missing** | **Method** | $AP_{ped.}$ | $AP_{div.}$ | $AP_{bou.}$ | **mAP** |
| Front View (Center) | MapTR | 25.7 | 34.5 | 33.6 | 31.3 |
|  | **Ours** | **36.6** | **45.0** | **45.8** | **42.4** |
| Front Left View (Left) | MapTR | 37.9 | 47.8 | 45.6 | 43.7 |
|  | **Ours** | **44.3** | **52.0** | **52.4** | **49.5** |
| Front Right View (Right) | MapTR | 38.8 | 46.6 | 47.1 | 44.2 |
|  | **Ours** | **45.3** | **52.3** | **51.7** | **49.4** |
| Back View (Center) | MapTR | 33.4 | 25.2 | 27.0 | 28.5 |
|  | **Ours** | **39.6** | **40.8** | **41.2** | **40.5** |
| Back Left View (Left) | MapTR | 41.3 | 48.3 | 47.8 | 45.8 |
|  | **Ours** | **45.5** | **53.1** | **52.9** | **50.5** |
| Back Right View (Right) | MapTR | 41.2 | 49.5 | 47.8 | 46.1 |
|  | **Ours** | **46.4** | **53.1** | **53.2** | **50.9** |

the X/Y axes, respectively.

**Implementation Details.** Our proposed SafeMap framework is trained using four NVIDIA RTX 3090 GPUs. To simulate real scenarios, we randomly discard RGB images of any view in each sample during training, reflecting the loss of frames from that view in actual situations. We select two baseline models, MapTR (Liao et al., 2023a) and HIMap (Zhou et al., 2024), and retrain them using their configurations. All experiments utilize the AdamW optimizer (Loshchilov & Hutter, 2019) with a learning rate of $4.2 \times 10^{-4}$, fine-tuning for 8 epochs, while hyperparameters $\lambda_1$ and $\lambda_2$ are set to 0.05 and 5, respectively.

The original images have a resolution of $1,600 \times 900$, resized by a factor of 0.5 during training. We limit the maximum number of map elements per frame to 100, with each containing 20 points. The size of each Bird's-Eye-View (BEV) grid is set to 0.75 meters, and the transformer decoder is configured with two layers. For the G-PVR module, the SafeMap model undergoes fine-tuning for 8 epochs on the nuScenes dataset and 2 epochs on the Argoverse2 dataset, with batch sizes set to 4 and 6, respectively. During inference, we evaluate the model across possible scenarios.

### 4.2. Comparisons with State-of-the-Art Methods

**Results on nuScenes.** Tab. 1 and Tab. 2 present the results of our SafeMap model compared to the representative MapTR (Liao et al., 2023a) and HIMap (Zhou et al., 2024) methods in the HD map construction task under both complete and incomplete observations on the nuScenes dataset. For fairness, we used the official model configurations provided by the open-source codebases and retrained the mod-

*Table 2.* Performance comparisons with (Zhou et al., 2024) when losing each of six camera views on the nuScenes validation set.

| Standard | Method | $AP_{ped.}$ | $AP_{div.}$ | $AP_{bou.}$ | mAP |
|---|---|---|---|---|---|
| All Views | HIMap | 62.2 | 66.5 | 67.9 | 65.5 |
|  | **Ours** | **62.6** | **66.7** | **68.7** | **66.0** |

| View Missing | Method | $AP_{ped.}$ | $AP_{div.}$ | $AP_{bou.}$ | mAP |
|---|---|---|---|---|---|
| Front View (Center) | HIMap | 39.2 | 41.6 | 33.1 | 38.0 |
|  | **Ours** | **50.7** | **55.7** | **56.3** | **54.3** |
| Front Left View (Left) | HIMap | 51.5 | 59.6 | 60.0 | 57.0 |
|  | **Ours** | **59.2** | **64.2** | **65.7** | **63.0** |
| Front Right View (Right) | HIMap | 57.0 | 62.1 | 62.6 | 60.6 |
|  | **Ours** | **60.0** | **64.2** | **66.1** | **63.4** |
| Back View (Center) | HIMap | 46.3 | 31.4 | 21.7 | 33.1 |
|  | **Ours** | **51.4** | **50.8** | **51.6** | **51.3** |
| Back Left View (Left) | HIMap | 58.1 | 63.8 | 64.2 | 62.0 |
|  | **Ours** | **60.7** | **65.4** | **67.0** | **64.4** |
| Back Right View (Right) | HIMap | 58.5 | 62.8 | 63.4 | 61.6 |
|  | **Ours** | **60.8** | **65.1** | **66.5** | **64.2** |

*Table 3.* Performance comparison on MapTR (Liao et al., 2023a) when losing each of seven camera views on Argoverse2 val set.

| Standard | Method | $AP_{ped.}$ | $AP_{div.}$ | $AP_{bou.}$ | mAP |
|---|---|---|---|---|---|
| All Views | MapTR | 57.7 | 58.9 | 59.4 | 58.7 |
|  | **Ours** | **58.7** | **59.7** | **60.6** | **59.7** |

| View Missing | Method | $AP_{ped.}$ | $AP_{div.}$ | $AP_{bou.}$ | mAP |
|---|---|---|---|---|---|
| Front View (Center) | MapTR | 50.8 | 49.8 | 53.9 | 51.5 |
|  | **Ours** | **53.8** | **54.9** | **58.1** | **55.6** |
| Front Left View (Left) | MapTR | 51.8 | 55.4 | 53.4 | 53.5 |
|  | **Ours** | **54.6** | **58.6** | **57.9** | **57.0** |
| Front Right View (Right) | MapTR | 52.7 | 57.3 | 54.2 | 54.7 |
|  | **Ours** | **55.6** | **58.9** | **57.3** | **57.3** |
| Rear Left View (Left) | MapTR | 50.5 | 48.1 | 50.0 | 49.5 |
|  | **Ours** | **54.6** | **53.9** | **55.5** | **54.7** |
| Rear Right View (Right) | MapTR | 49.1 | 52.6 | 47.0 | 49.6 |
|  | **Ours** | **54.8** | **57.2** | **54.0** | **55.3** |
| Side Left View (Left) | MapTR | 55.0 | 57.9 | 57.2 | 56.7 |
|  | **Ours** | **57.4** | **59.5** | **59.7** | **58.9** |
| Side Right View (Right) | MapTR | 56.0 | 58.3 | 57.8 | 57.4 |
|  | **Ours** | **58.0** | **59.3** | **59.8** | **59.1** |

els according to their default settings. The experimental results reveal the following: **1)** Both MapTR and HIMap models perform poorly when the 'CAM FRONT' and 'CAM BACK' views are missing. These views have the most significant impact on model performance due to their inclusion of more critical map elements. **2)** The SafeMap model consistently outperforms the source MapTR/HIMap models in both complete and incomplete observation scenarios. Notably, SafeMap improves the HIMap model's mAP metric by 2.4% to 18.2% across various missing camera view cases, demonstrating its superior generalization capability in HD map construction. **3)** SafeMap can be seamlessly integrated into existing models, offering a plug-and-play solution for enhanced robustness. Its excellent performance under incomplete view conditions is attributed to its comprehensive feature reconstruction capability, which effectively restores missing view features from incomplete data.

**Results on Argoverse2.** Tab. 3 compares the performance of our SafeMap model with the popular MapTR(Liao et al., 2023a) model on the Argoverse2 dataset. Compared to the nuScenes dataset, both SafeMap and MapTR exhibit less performance degradation under incomplete view conditions. This is likely due to the Argoverse2 dataset capturing observations from seven viewpoints, whereas the nuScenes dataset only has six. In the complete view observation setting, our SafeMap model achieves a 1.0% absolute improvement in mAP compared to the source MapTR model. Under incomplete observations, SafeMap significantly outperforms the baseline MapTR model across various missing camera view scenarios. For instance, when the front camera view is missing, SafeMap achieves a 4.1% improvement in mAP over MapTR. These results further demonstrate the

*Table 4.* Ablation study of components on the Gaussian-based Perspective View Reconstruction module (G-PVR) and the Distill-based BEV Correction module (D-BEVC).

| G-PVR | D-BEVC | $AP_{ped.}$ | $AP_{div.}$ | $AP_{bou.}$ | mAP |
|---|---|---|---|---|---|
| ✗ | ✗ | 36.4 | 42.0 | 41.5 | 39.9 |
| ✓ | ✗ | 42.7 | 47.7 | 49.2 | 46.5 |
| ✗ | ✓ | 42.4 | 47.7 | 49.4 | 46.5 |
| ✓ | ✓ | **42.9** | **49.4** | **49.5** | **47.3** |

generalization ability of our method across different sensor configurations in the HD map construction task. Overall, SafeMap significantly outperforms previous methods across both complete and incomplete observations, showcasing the benefits of the G-PVR and D-BEVC modules.

### 4.3. Ablation Study

In this section, we analyze the SafeMap model to verify the effectiveness of the proposed method. Unless otherwise specified, all experiments use MapTR as the baseline model on the nuScenes dataset. For brevity, we report average metrics across six missing camera view cases in the ablation experiments.

**Analysis on Different Modules.** To systematically evaluate the contribution of each module in SafeMap, we conducted ablation studies by removing components individually and presenting the results in Tab. 4. The following ablation models were designed: **1)** SafeMap (Baseline): the model trained without the reconstruction module; **2)** SafeMap (w/

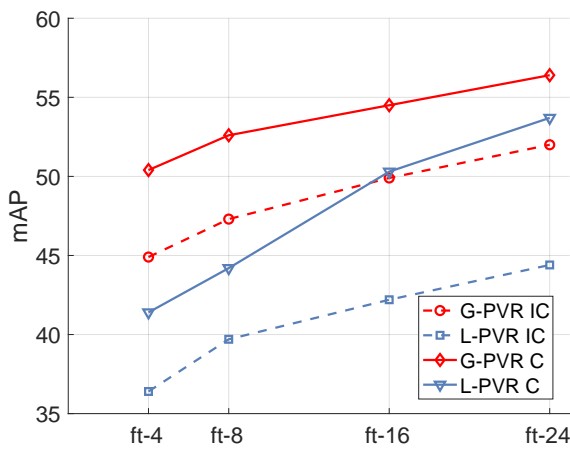

*Figure 4.* Comparisons of the Gaussian-based PVR *vs.* Local PVR. ft-$N$ denotes fine-tuning $N$ epochs.

*Table 5.* Ablation study on the use of the G-PVR module.

| Setting | $\text{AP}_{ped.}$ | $\text{AP}_{div.}$ | $\text{AP}_{bou.}$ | mAP |
|---|---|---|---|---|
| SafeMap (*w/o* G-PVR) | 42.4 | 47.7 | 49.4 | 46.5 |
| SafeMap (*w/* Mean-PVR) | 35.8 | 41.0 | 42.8 | 39.9 |
| SafeMap (*w/* MAE-PVR) | 42.3 | 47.9 | 49.2 | 46.5 |
| SafeMap (*w/* Standard-PVR) | 42.6 | 48.9 | 48.7 | 46.8 |
| **SafeMap (*w/* Gaussian-PVR)** | **42.9** | **49.4** | **49.5** | **47.3** |

*Table 6.* Ablation study on the use of the distillation loss.

| Method | $\text{AP}_{ped.}$ | $\text{AP}_{div.}$ | $\text{AP}_{bou.}$ | mAP |
|---|---|---|---|---|
| SafeMap (*w/o* D-BEVC) | 42.7 | 47.7 | 49.2 | 46.5 |
| SafeMap (*w/* $KL$) | 42.3 | 48.6 | 49.5 | 46.8 |
| SafeMap (*w/* $L_1$) | 42.8 | 49.2 | 49.4 | 47.1 |
| **SafeMap (*w/* $L_2$)** | **42.9** | **49.4** | **49.5** | **47.3** |

G-PVR): the model trained with the Gaussian-based Perspective View Reconstruction (G-PVR) module; **3)** SafeMap (w/ D-BEVC): the model trained with the Distill-based BEV Correction (D-BEVC) module; **4)** SafeMap with both G-PVR and D-BEVC modules (the default setting). As shown in Tab. 4, the results demonstrate that SafeMap models with the G-PVR and D-BEVC modules significantly outperform the baseline, verifying the effectiveness of these modules in reconstructing missing camera view features. The ablation study confirms that each module in SafeMap meaningfully contributes to improving performance in settings with incomplete observations.

**Gaussian-based PVR *vs.* Local PVR.** To explore the impact of different Perspective View Reconstruction strategies, we evaluated Gaussian-based and Local-level reconstruction modules and presented the results in Fig. 4. The strategies are as follows: **1)** Gaussian-based PVR: uses all available views as context to reconstruct missing view information by leveraging relationships among camera views. **2)** Local PVR: follows the approach of M-BEV (Chen et al., 2024b), using context from neighboring views to recover missing views. As shown in Fig. 4, the Gaussian-based PVR method consistently outperforms the Local PVR method in all settings, including both complete and incomplete observations. The superior performance of the Gaussian-based PVR module is attributed to its ability to leverage all views for reconstructing missing map elements, which is crucial for enabling the ego-vehicle to accurately locate itself and anticipate upcoming features.

**Ablation Study on G-PVR Module.** To systematically evaluate the effectiveness of the G-PVR module, we trained the model with various variants and reported the mAP results in Tab. 5. The variants include Mean-PVR, MAE-PVR, Standard-PVR, and G-PVR. Each variant employs a different approach to reconstruct the missing view. Mean-PVR calculates the mean of all available views, MAE-PVR uses all views to reconstruct the missing one with a masked token, and Standard-PVR utilizes a deformable attention module to aggregate information from available views.

As shown in Tab. 5, the experimental results reveal the following: **1)** The Mean-PVR variant underperformed compared to the baseline model, indicating that simple PVR methods are insufficient for reconstructing missing view information. **2)** The G-PVR module consistently outperforms both the MAE-PVR and Standard-PVR variants. This performance is attributed to G-PVR's use of Gaussian-based sparse sampling to generate strategic reference points, which serve as spatial priors for missing views. By leveraging these reference points, deformable attention focuses on the most informative regions across all available views, facilitating efficient reconstruction of the missing data. These findings confirm that the G-PVR module effectively utilizes valuable information in the remaining complete views to reconstruct the missing view information.

**Ablation Study on Distillation Loss.** We also investigated the impact of different distillation loss functions within the D-BEVC module. The evaluated loss functions include Manhattan distance ($L_1$), Euclidean distance ($L_2$), and Kullback-Leibler Divergence ($KL$). As shown in Tab. 6, SafeMap utilizing the $KL$ divergence demonstrated inferior performance compared to both $L_1$ and $L_2$. The experimental results indicate that SafeMap with $L_2$ achieves the best performance, which explains why $L_2$ was selected as the default configuration in our experiments.

**Impact of Different Numbers of Missing Views.** To investigate the impact of varying numbers of missing camera views on HD map construction, we randomly zeroed out 1, 2, 3, 4, and 5 camera images in the nuScenes dataset. Each configuration yields several combinations of missing views; specifically, there are 6 combinations for 1 missing

*Table 7.* Impact of different numbers of missing views.

| Method | #View | $\text{AP}_{ped.}$ | $\text{AP}_{div.}$ | $\text{AP}_{bou.}$ | mAP |
|---|---|---|---|---|---|
| | 1× | 36.4 | 42.0 | 41.5 | 39.9 |
| | 2× | 27.5 | 31.3 | 29.9 | 29.6 |
| MapTR | 3× | 18.8 | 20.9 | 18.8 | 19.5 |
| | 4× | 11.0 | 11.6 | 9.4 | 10.6 |
| | 5× | 4.6 | 4.3 | 3.0 | 4.0 |
| | 1× | 42.9 | 49.4 | 49.5 | 47.3 |
| | 2× | 33.5 | 37.0 | 35.0 | 35.2 |
| **SafeMap** | 3× | 23.7 | 24.4 | 20.2 | 22.8 |
| | 4× | 15.6 | 16.2 | 12.6 | 14.8 |
| | 5× | 6.7 | 7.0 | 4.3 | 6.0 |

*Table 8.* Accuracy-computation analysis. We report the mAP performance under the "complete" / "incomplete" observations.

| Method | mAP | GPU Mem | Param | FPS |
|---|---|---|---|---|
| MapTR | 50.3 / 39.9 | 2298 MB | 39.1 M | 21.5 |
| **SafeMap** | **52.5 / 47.3** | **2300 MB** | **39.5 M** | **21.4** |
| HIMap | 65.5 / 52.1 | 4091 MB | 68.1 M | 9.7 |
| **SafeMap** | **66.0 / 60.1** | **4155 MB** | **71.7 M** | **9.2** |

view, 15 for 2, 20 for 3, 15 for 4, and 6 for 5. We compute the average AP and mAP metrics for each combination to derive the final results. To demonstrate the robustness of our model, we trained it to handle all these cases. As shown in Tab. 7, the results reveal that as the number of missing views increases, model performance declines, as expected. Compared to the MapTR (Liao et al., 2023a) model, SafeMap shows significantly less performance degradation with missing perspectives. This resilience stems from the ability of SafeMap to effectively reconstruct features from incomplete views, enhancing robustness despite multiple missing cameras.

**Inference Speed, Model Size & GPU Memory.** To evaluate the effectiveness of SafeMap, we analyzed its performance in terms of accuracy, model size, GPU memory usage, and inference speed. The results in Tab. 8 reveal several key findings: **1)** SafeMap significantly outperforms the source MapTR/HIMap models in both complete and incomplete observations. **2)** In terms of model size, SafeMap only increases the number of parameters by 0.4MB to 3.6MB compared to the source models. **3)** Regarding GPU memory and inference speed, SafeMap and the source models show nearly identical metrics. Overall, SafeMap achieves substantial performance improvements over the baseline models with minimal increases in parameters, while maintaining comparable inference speed and memory usage.

**Robustness against Camera Sensor Corruptions.** To further evaluate the robustness of SafeMap, we assessed its per-

*Table 9.* Experimental results on the robustness of HD map construction under camera sensor corruptions.

| Method | $\text{AP}_{ped.}$ | $\text{AP}_{div.}$ | $\text{AP}_{bou.}$ | mAP | mRR↑ | mCE↓ |
|---|---|---|---|---|---|---|
| MapTR | 46.3 | 51.5 | 53.1 | 50.3 | 49.3 | 100.0 |
| **SafeMap** | **48.1** | **54.3** | **55.3** | **52.5** | **51.2** | **90.6** |
| HIMap | 62.2 | 66.5 | 67.9 | 65.5 | 56.6 | 100.0 |
| **SafeMap** | **62.6** | **66.7** | **68.7** | **66.0** | **62.8** | **83.2** |

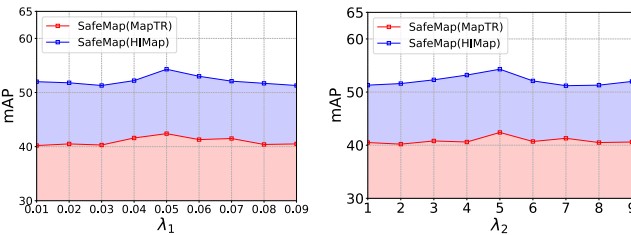

*Figure 5.* **Sensitivity Analysis of Hyperparameters $\lambda_1$ and $\lambda_2$.**

formance against eight real-world camera sensor corruptions from MapBench (Hao et al., 2024a), categorized into exterior environments, interior sensors, and sensor failures. The details of these corruptions are outlined in MapBench (Hao et al., 2024a). As shown in Tab. 9, SafeMap achieves gains of 1.9% in mRR and 9.4% in mCE compared to the baseline MapTR model. It also shows improvements of 6.2% in mRR and 16.8% in mCE over the original HIMap model. These results underscore SafeMap's effectiveness in enhancing the robustness of camera-based HD map construction methods against various sensor corruptions.

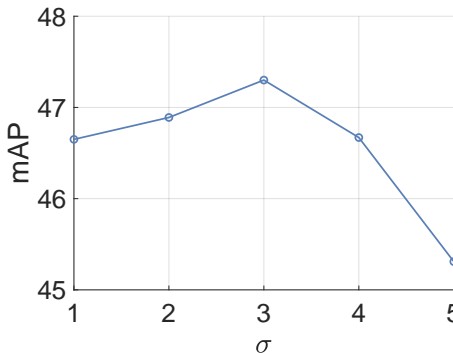

*Figure 6.* **Sensitivity Analysis of Hyperparameter Variance $\sigma$.**

### 4.4. Sensitivity of Hyper-parameters

**Sensitivity Analysis of Hyperparameters $\lambda_1$ and $\lambda_2$.** We analyze the sensitivity of hyperparameters $\lambda_1$ and $\lambda_2$, as illustrated in Fig. 5. The results presented focus on the miss Front View setting for the SafeMap (MapTR) and SafeMap (HIMap) methods on the nuScenes dataset. By varying $\lambda_1$ and $\lambda_2$ within a feasible range while keeping other hyperparameters at their default values, we gain insights into each component's contribution to overall performance. As

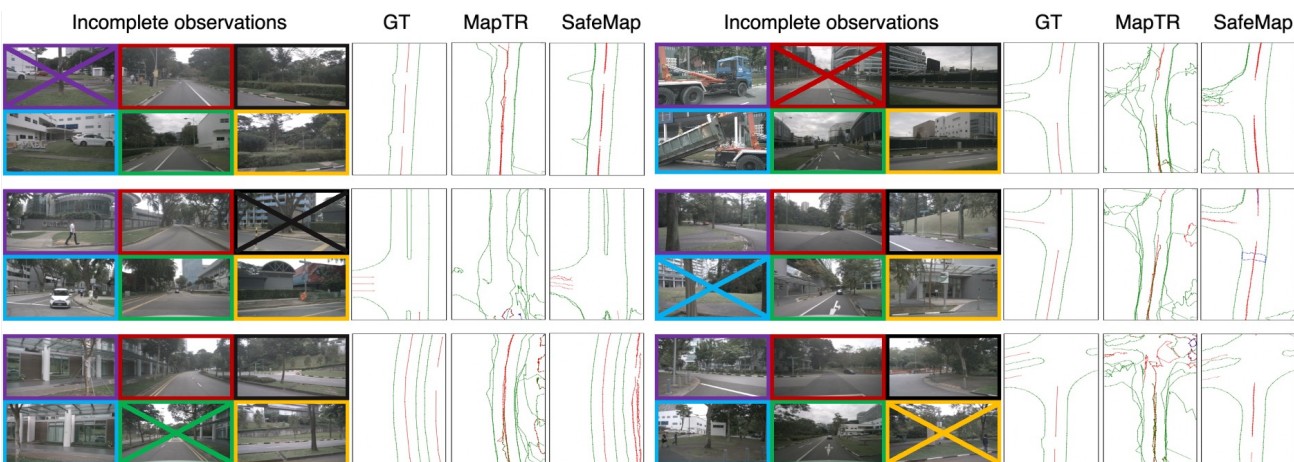

*Figure 7.* Qualitative Comparisons. The camera view marked with the symbol ✕ indicates the absence of this perspective.

shown, progressively increasing $\lambda_1$ from 0.01 to 0.09 leads to consistent improvements, achieving optimal performance at $\lambda_1 = 0.05$. Additionally, Fig. 5 demonstrates that the best performance for our method occurs at $\lambda_2 = 5$. Therefore, we set $\lambda_1$ to 0.05 and $\lambda_2$ to 5 for all configurations.

**Sensitivity Analysis of Hyperparameter Variance $\sigma$ in Eq. 2.** We conduct experiments using the results from the 'miss all view' setting for the SafeMap (MapTR) model on the nuScenes dataset. The sensitivity of hyperparameter variance $\sigma$ in Eq. 2 is illustrated in Fig. 6. Across a wide range of $\alpha$ in $[1, 5]$, as we progressively increase the threshold $\sigma$ in Eq. 2 from 1 to 5, our method consistently performs well, achieving the best results at $\sigma = 3$. Therefore, we set $\sigma$ to 3 for all configurations.

### 4.5. Qualitative Results

We present the online vectorized HD map construction results for both the MapTR and SafeMap models under conditions of incomplete observations, as shown in Fig. 7. Our analysis reveals that the absence of different perspective images affects map construction to varying degrees. Notably, the lack of front and back perspective images has a significant negative impact, while the absence of other perspectives exerts a relatively minor influence. This finding is consistent with the results shown in Tab. 1 and Tab. 2, as front and back observations provide crucial information relevant to map elements. In comparison to the baseline MapTR model, SafeMap exhibits a significant advantage in effectively modeling all map elements under conditions of incomplete observations. Overall, our model shows significant advantages in the incomplete observations setting.

## 5. Conclusion

In this paper, we present SafeMap, a novel robust framework for high-definition (HD) map construction that ensures accuracy in the presence of missing camera views. At the core of SafeMap is the Gaussian-based Global View Reconstruction (G-GVR) module, which effectively leverages relationships among available camera views to infer missing information. Additionally, SafeMap incorporates a Distillation-based Bird's-Eye-view Correction (D-BEVC) module, which uses the complete panoramic BEV features to further correct the BEV features extracted from incomplete observations. Extensive experimental evaluations validate that SafeMap significantly outperforms baseline models across both complete and incomplete observation scenarios, achieving substantial performance gains while maintaining comparable inference speed and memory efficiency.

## Impact Statement

While SafeMap achieves impressive results in high-definition map construction with missing camera views, it does not address the challenges of multi-sensor fusion. This limitation could impact its effectiveness in real-world applications where data from multiple sensor modalities is critical. Future research should explore the development of multi-modal robust architectures that can effectively integrate information from various sensor types, enhancing the overall resilience and accuracy of map construction in diverse environments. As far as we know, our work does not have any negative ethical impacts or concerning societal consequences, as it focuses on advancing fundamental research in autonomous driving robustness.

**Acknowledgment:** This project has received funding from the NSF China (No. 62302139, No.62106259), and FRFCU China(No. PA2025IISL0113).

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
