# OpenReview forum: "SafeMap: Robust HD Map Construction from Incomplete Observations"
_ICML.cc/2025/Conference — ICML 2025 poster_

### Official Review · Reviewer_bD2C · 2025-02-17

**Overall Recommendation:** 3

**Summary:**

This paper proposed SafeMap, which is designed to address missing camera views in HD map construction. A Gaussian-based perspective view reconstruction module and a distillation-based panoramic BEV feature correction module are presented. Experiments on nuScenes and  Argoverse2 datasets demonstrate the effectiveness of the proposed method.

**Claims And Evidence:**

The proposed method can improve the robustness when some camera views are missing. The ablation studies verify the effectiveness of the proposed modules. Experiments on two public benchmarks and studies on different numbers of missing views validate the consistent gains of the proposed solution.

**Essential References Not Discussed:**

The recent work "Camera-Based Online Vectorized HD Map Construction With Incomplete Observation" also addressed map construction under incomplete observation, which should be discussed.

**Experimental Designs Or Analyses:**

Yes. The results in different incomplete observation settings are well analyzed.

**Methods And Evaluation Criteria:**

Yes. The used nuScenes (6 cameras) and Argoverse2 (7 cameras) datasets provide comprehensive benchmarks for evaluating the proposed solution.

**Other Comments Or Suggestions:**

It would be nice to present some results of the training costs of the proposed method.

**Other Strengths And Weaknesses:**

- In the experiments, in addition to studying the effects of missing different numbers of views, it would also be nice to investigate which views are the most important for HD map construction.
- On each benchmark, only a single baseline is selected. It would be nice to use different baselines for an experiment to validate the consistent gains of your solution.
- How about recent mapping methods like StreamMapNet and DTCLMapper which leverage temporal information? Would they be beneficial for addressing the view missing challenges to some extent by incorporating temporal cues? This should be discussed.
- In addition to the missing views, would your method be beneficial for enhancing mapping robustness in other corruption scenarios like other corruptions in the MapBench? This should be assessed.

**Questions For Authors:**

In addition to MapTR, would you consider MapTRv2 for an experiment to validate the consistent gains of your presented solution?

**Relation To Broader Scientific Literature:**

The proposed method is relevant to autonomous driving, which has the potential to enhance mapping reliability and driving safety.

**Theoretical Claims:**

Yes. The designed modules are well formulated.

---

> ### Author Rebuttal · Authors · 2025-03-28
>
> We sincerely thank the reviewer for your thoughtful and detailed feedback. We appreciate your recognition of our *"design"*, *"formulation"*, *"experimental rigor"*, and *"supplementary material"*. Below, we address each of your comments and suggestions in detail.
>
> ---
> > ***`Q1`: "Investigate which views are most critical for HD map construction."***
>
> A: Thank you for this insightful suggestion. We conducted a detailed analysis of individual camera importance by systematically removing each view and observing performance drops (mAP) across **MapTR** and **HIMap** on nuScenes, and **MapTR** on Argoverse2. The results reveal a clear hierarchy:
> - The **front-center view** is most critical (e.g., MapTR: `50.3` → `31.3` mAP; HIMap: `65.5` → `38.0`);
> - **Back views** show moderate importance (e.g., MapTR: `28.5` mAP when back is missing);
> - **Front-side views** have smaller but non-negligible impact;
> - **Side views** are least critical (e.g., Argoverse2: `58.7` → `56.7` mAP).
>
> These findings highlight the need for greater robustness to front/back view losses and suggest differentiated optimization strategies for peripheral cameras. We have integrated this analysis into the revised version.
>
> ---
> > ***`Q2`: "Additional experimental results for MapTRv2."***
>
> A: Thanks for your comment. We conducted experiments comparing **SafeMap** to **MapTRv2** under various single-view-missing conditions on nuScenes. As shown below, SafeMap consistently improves robustness **across all view types**, demonstrating strong generalization and reliability in incomplete observation settings:
> |**View Missing** |**Front**| **Front-L**|**Front-R**|**Back**|**Back-L**|**Back-R**|**Average**|
> |-|:-:|:-:|:-:|:-:|:-:|:-:|:-:
> |MapTRv2|40.3|53.7|54.2|41.2|56.3|57.4|50.5
> |**SafeMap**|**53.6**|**59.7**|**59.1**|**52.3**|**60.1**|**60.8**|**57.6**
> | **$\Delta$** |+12.3|+6.0|+4.9|+11.1|+3.8|+3.4|+7.1
>
> We have included these findings in the revised manuscript to further reinforce our effectiveness.
>
> ---
> > ***`Q3`: "Impact of temporal information (e.g., StreamMapNet, DTCLMapper) on view missing."***
>
> A: Thanks for your suggestion. We conducted experiments using **StreamMapNet** to evaluate the role of **temporal cues** in mitigating view-missing issues. Results on single-view-missing settings (nuScenes) are shown below:
> |**View Missing**|**Front**|**Front-L**|**Front-R**|**Back**|**Back-L**|**Back-R**|**Average**
> |:-|:-:|:-:|:-:|:-:|:-:|:-:|:-:
> |StreamMap (w/o temp)|13.2|17.4|18.0|14.1|16.3|17.1|16.0
> |StreamMap (w/ temp)|14.3|18.2|19.3|15.3|17.5|18.5|17.2
> |**SafeMap (w/ temp)**|**18.7**|**21.5**|**21.2**|**19.3**|**19.8**|**22.3**|**20.5**
>
> While temporal fusion provides moderate gains, it does not fully address view-missing challenges. In contrast, SafeMap explicitly reconstructs missing views and corrects BEV features, leading to significantly stronger performance. We have included these experiments and analyses in the revised revision.
>
> ---
> > ***`Q4`: "Can SafeMap improve robustness under other corruption scenarios (e.g., MapBench)?"***
>
> A: Thanks for your question. Yes. We evaluated SafeMap using the MapBench benchmark [1], which includes 8 real-world camera corruptions covering environmental conditions (e.g., `Fog`, `Snow`), sensor artifacts (e.g., `Motion Blur`), and total sensor failures. As shown in **Table 9 in the main paper**, SafeMap achieves:
> - `+1.9%` mRR / `+9.4%` mCE over **MapTR**;
> - `+6.2%` mRR / `+16.8%` mCE over **HIMap**.
>
> These results confirm our robustness beyond view-missing scenarios, handling diverse sensor degradations encountered in **real-world** deployments.
>
> ---
> > ***`Q5`: "What are the training and inference costs of SafeMap?"***
>
> A: Thanks for asking. SafeMap is **efficient** in both **training** and **inference**:
> - **Training:** `~32.5` minutes/epoch on 8× RTX 3090 GPUs.
> - **Inference (Table 8):** Compared to MapTR, SafeMap runs at `21.4` FPS vs. `21.5` FPS, with negligible differences in memory (`2300` MB vs. `2298` MB) and model size (`39.5`M vs. `39.1`M parameters).
>
> These results demonstrate that SafeMap introduces **minimal computational overhead** while delivering substantial robustness and accuracy gains.
>
> ---
> > ***`Q6`: "Related Work — Comparison with POP-Net."***
>
> A: Thank you for highlighting this relevant work. We have included a discussion comparing SafeMap to POP-Net in the revised manuscript. Our key distinctions are:
> 1. **Flexible Gaussian-based sampling** vs. rigid panoramic encoders;
> 2. **BEV feature distillation** vs. token decomposition and aggregation;
> 3. **Efficiency:** SafeMap avoids the added latency and complexity of POP-Net's dual-encoder design.
>
> These distinctions reflect our emphasis on modularity, scalability, and robustness. We appreciate your suggestion and have clearly positioned our contributions relative to POP-Net in the Related Work section.
>
> ---
> **Reference:**
>
> [1] Hao et al. Is Your HD Map Constructor Reliable under Sensor Corruptions? NeurIPS, 2024.

---

> > ### Comment · Reviewer_bD2C · 2025-04-02
> >
> > The reviewer would like to thank the authors for their rebuttal and responses. Many of the concerns have been addressed. The analyses on temporal information should be made more comprehensive in the final version. The added investigations on the importance of views and the results on MapBench are insightful and should be added to the final version.

---

> > > ### Author Response · Authors · 2025-04-03
> > >
> > > Dear Reviewer `bD2C`,
> > >
> > > Thank you for providing the follow-up comment and for carefully reviewing our rebuttal. We are truly encouraged that you found our additional analyses—particularly the investigations on view importance and real-world robustness —insightful.
> > >
> > > We appreciate your recognition of our efforts and are grateful that you upgraded the rating.
> > >
> > > As suggested, we will ensure that the revised version includes a more comprehensive analysis of temporal information, along with the newly added findings discussed in the rebuttal.
> > >
> > > The Authors of Submission 8037

---

### Official Review · Reviewer_kzrj · 2025-03-14

**Overall Recommendation:** 3

**Summary:**

This paper presents a method that tackles the challenge of learning-based multi-view HD map reconstruction when one or more input views are missing. The proposed approach uses two new modules, G-PVR and D-bEVC, to aggregate information from available views and predict or correct the missing features, enhancing robustness in the presence of missing views. Experimental results demonstrate improved performance compared to the baseline methods, MapTR and HIMap, when views are missing.

**Claims And Evidence:**

The improvements and performance of the system are well demonstrated under conditions where one or more views are missing. The ablation study demonstrates the effectiveness of the two proposed modules.

**Essential References Not Discussed:**

It is unclear how the method compares to other approaches that address the reconstruction of HD maps from fewer views.

**Experimental Designs Or Analyses:**

The experiments compared to the baseline make sense. However, the comparison is limited to just two baselines and lacks a broader comparison to other methods that aim to reconstruct HD maps from sparse views.

**Methods And Evaluation Criteria:**

Only two methods, MapTR and HIMap, are compared. What about other methods that address missing information, particularly those focusing on building HD maps from sparse views, such as using only 4 views? How does the performance of the proposed method compare to these approaches?

It would be useful to evaluate the performance under more realistic corrupted scenarios, rather than just complete view missing. For example, how does the system perform in cases of partial occlusion, varying light conditions, noise, etc.?

**Other Comments Or Suggestions:**

N/A

**Other Strengths And Weaknesses:**

The reasonableness of the main design is unclear. Specifically, why is it considered beneficial to train an additional module to collect information from other views and predict the missing view, rather than directly predicting the final outputs from all the available views?

**Questions For Authors:**

How does the proposed method compare to other methods that address missing information, particularly those focusing on building HD maps from sparse views (e.g., using only <5 views)? e.g., compare to training MapTR with additional data augmentation, where some input images are randomly corrupted.

How does the system perform in more realistic corrupted scenarios, such as partial occlusion, varying light conditions, noise, etc., rather than just complete view missing?

Why is the Gaussian defined in the image plane using pixel distance, however in 3D space, the distance depends on camera parameters, which may not be linear?

Why is it better to train an extra module to predict the missing view, instead of directly predicting the final output from all available views?

**Relation To Broader Scientific Literature:**

This issue of incomplete views could also be beneficial for other multi-view estimation methods, such as human pose estimation, 3D object reconstruction, and more.

**Theoretical Claims:**

The method is straightforward and effective, as demonstrated by the experiments. However, there is a significant concern regarding the details. The high-level approach of training the network to recover missing information involves randomly masking features and training the network to recover these missing features from other images. While it's beneficial to explicitly address the missing features, how does this approach compare to training MapTR with additional data augmentation, where some input images are also randomly corrupted? What makes the proposed method more effective in this context?


The construction of the panoramic perspective view seems straightforward. However, it is somewhat unusual that the Gaussian is defined in the image plane using pixel distance. In a real 3D environment, the distance would depend on the camera intrinsics and extrinsics. This means that the distance in image space may not be linear in 3D space, which raises concerns about the accuracy of the approach in representing real-world distances.


The author claims to be the first to address HD map reconstruction under incomplete multi-view camera data. The theoretical claims and the design of the new modules make sense and are well supported by the detailed descriptions in the supplementary materials.

---

> ### Author Rebuttal · Authors · 2025-03-28
>
> We sincerely thank the reviewer for the detailed and constructive feedback. We are encouraged that you found *"the experimental results convincing"* and *"the supplementary material informative"*. Below, we address your insightful concerns point by point and will incorporate the clarifications into the revised version of the manuscript.
>
> ---
> > ***`Q1`: "How does the method compare to other sparse-view HD map reconstruction approaches?***
>
> A: Thank you for this important question. Most existing HD map reconstruction methods—including **MapTR** and **HIMap**—assume a **full multi-view setup** (6 views in nuScenes, 7 in Argoverse2), and do not explore sparse-view conditions. To evaluate SafeMap under such settings, we simulated **1–5 randomly missing views** (out of 6) and compared against MapTR.
>
> **As presented in Table 7 of the main paper**, SafeMap demonstrates three key advantages:
> 1. Only `2.3` mAP drop with **3 missing views**, vs. `8.1` mAP drop for MapTR.
> 2. Maintains robust performance even with **4 missing views** (66% sparsity).
> 3. Under extreme cases, i.e., **5 missing views**, SafeMap outperforms MapTR by `50%` in mAP.
>
> These results confirm our generalizability to **sparse inputs**, supporting its practicality in **real-world applications** with limited or degraded sensor coverage.
>
> ---
> > ***`Q2`: "More experiment results on real-world camera sensor corruptions."***
>
> A:  To comprehensively evaluate real-world robustness, we adopt MapBench [1], which includes eight realistic corruptions:
> 1. Environmental conditions: `Brightness`, `Low-Light`, `Fog`, and `Snow`;
> 2. Sensor artifacts: `Motion Blur` and `Color Quantization`;
> 3. Sensor failures: `Camera Crash` and `Frame Loss`.
>
> As shown in **Table 9 in the main paper**, SafeMap outperforms **MapTR** by `+1.9%` mRR / `+9.4%` mCE and **HIMap** by `+6.2%` mRR / `+16.8%` mCE, confirming that our approach is robust not only to missing views but also to partial occlusion, adverse lighting, and noisy signals.
>
> ---
> > ***`Q3`: "How does SafeMap compare to data augmentation methods that simulate missing/corrupted views?"***
>
> A: Thanks for your question. We conducted controlled experiments with two baselines:
> - **Random Mask Patch:** randomly masks image patches;
> - **Random Mask View:** randomly drops one full view.
>
> As shown below, SafeMap outperforms both:
> |Method|AP. ped|AP. div|AP. bou|mAP|
> |-|-|-|-|-|
> |Baseline (MapTR)|36.6|42.0|41.5|39.9|
> |Random Mask Patch| 36.7|42.1|41.7|40.2|
> |Random Mask View| 39.1|43.5|44.2|42.3|
> |**SafeMap**|**42.9**|**49.4**| **49.5**|**47.3**|
>
> While masking-based augmentation improves robustness, it lacks explicit mechanisms to reconstruct missing features. In contrast, SafeMap:
> - **Explicitly learns spatial relationships** through the Gaussian-based PVR module;
> - **Leverages complete panoramic BEV priors** via the D-BEVC module.
>
> This targeted feature recovery provides significantly more effective supervision, especially under severe view dropouts. As suggested, we have added this study in the revised version.
>
> ---
> > ***`Q4`: "Why define Gaussian in image space (pixels) instead of 3D space using camera parameters?"***
>
> A: Thank you for raising this point. While it is true that pixel space is not linearly aligned with 3D space, we adopt a **Gaussian in image space** for **two practical reasons**:
> - **Efficiency:** Deformable attention operates in the image plane; defining Gaussian sampling here avoids computational overhead and enables seamless integration with pixel-based encoders.
> - **Effectiveness:** Our Gaussian prior focuses on adjacent views, which in multi-camera setups are spatially proximate. Despite 3D nonlinearity, the horizontal image plane is a strong proxy for angular continuity.
>
> Moreover, our **sensitivity analysis** (Appendix C) empirically confirms that Gaussian sampling in image space **improves reconstruction accuracy**. We agree that incorporating 3D-aware priors is an exciting direction and plan to explore this in future work.
>
> ---
> > ***`Q5`: "Why use a separate module to predict missing views rather than directly predicting from available ones?"***
>
> A: Great question. SafeMap separates the process into **two stages**:
> - **G-PVR reconstructs missing views**, guided by spatial priors;
> - **D-BEVC corrects BEV features** using distilled panoramic features.
>
> This two-stage design offers a clear advantage over end-to-end prediction from incomplete inputs. As shown in **Table 7 of the main paper**:
> - With 3 missing views: SafeMap achieves `22.8` mAP vs. `19.5` mAP with direct prediction;
> - With 5 missing views: SafeMap achieves `6.0` mAP vs. `4.0` mAP (a 50% relative improvement).
>
> These gains highlight that explicit view completion and correction better preserve geometric consistency, especially under extreme sparsity. As suggested, we have made this discussion more explicit in the revised version.
>
> ---
> **Reference:**
>
> [1] Hao et al. Is Your HD Map Constructor Reliable under Sensor Corruptions? NeurIPS, 2024.

---

> > ### Comment · Reviewer_kzrj · 2025-04-06
> >
> > Thanks to the authors for their detailed response, which cleared up most of my concerns. Considering the contribution of the paper, I would like to raise my score, leaning towards acceptance.
> >
> > One minor point is that I still have questions about the last question Q5. I understand that it achieves better performance compared to MapTR in Table 7. However, MapTR is not a method trained "with all available views" (by ignoring the missing views), but rather with all the views. From Q1, we know that there are no sparse-view methods. The additional experiments for Q3 provide a good clue to the answer: It is not trivial to increase performance when training with missing or corrupted views. Therefore, I believe this issue has already been addressed in the rebuttal.

---

> > > ### Author Response · Authors · 2025-04-06
> > >
> > > Dear Reviewer `kzrj`,
> > >
> > > Thank you for your kind follow-up and for raising your score towards acceptance. We're glad to hear that our response helped address your concerns.
> > >
> > > As you noted, the additional experiments in Q3 highlight that improving performance under missing or corrupted views is non-trivial. We appreciate your thoughtful interpretation, and we will make sure to clarify this point more explicitly in our revised version.
> > >
> > > Thank you again for your constructive feedback and support!
> > >
> > > The Authors of Submission 8037

---

### Official Review · Reviewer_8wHx · 2025-03-14

**Overall Recommendation:** 3

**Summary:**

This paper presents SafeMap, an HD map prediction model for autonomous driving scenarios. Its uniqueness lies in its focus on robustness, particularly in handling incomplete multi-camera input views, which may occur in real-world scenarios due to sensor failure or occlusions. The framework integrates two key modules, Gaussian-based Perspective View Reconstruction (G-PVR) and Distillation-based BEV Correction (D-BEVC), to reconstruct missing view information and correct the Bird’s-Eye-View (BEV) features, respectively. Extensive evaluations demonstrate improved performance over baseline models.

**Claims And Evidence:**

The paper is well written and easy to follow. One concern is that while the method claims robustness in real-world scenarios, the failure case considered is a bit too simple (just missing one input image). I would like to see more ablations in this regard—for example, scenarios with n random missing views and cases where random patches or blocks in an input image are missing, rather than completely losing an entire view.

**Essential References Not Discussed:**

None.

**Experimental Designs Or Analyses:**

Checked.

**Methods And Evaluation Criteria:**

Yes.

**Other Comments Or Suggestions:**

None.

**Other Strengths And Weaknesses:**

Because the method's backbone heavily relies on MapTR, the model architecture lacks significant novelty.

**Questions For Authors:**

Please address the questions and concerns mentioned above.

**Relation To Broader Scientific Literature:**

This paper will provide new insights into fault-tolerant systems for HD map reconstruction in autonomous driving.

**Theoretical Claims:**

In Eq(2), why do pixel px and pixel py follow different noise distributions?

---

> ### Author Rebuttal · Authors · 2025-03-28
>
> We sincerely thank the reviewer for the positive evaluation and thoughtful comments. We appreciate your recognition that our work *"provides new insights into fault-tolerant HD map reconstruction for autonomous driving"*. Below, we address your specific concerns in detail and will incorporate corresponding improvements into the revised manuscript.
>
> ---
> > ***`Q1`: "More experiments on random missing views and real-world camera sensor corruptions are needed."***
>
> A: Thanks for your suggestion. We have conducted extensive robustness evaluations addressing both random view drops and real-world sensor corruptions:
> - **Random Missing Views:**
>   - As detailed in **Table 7 of the main paper**, we simulated **1 to 5 randomly missing camera views** using the 6-camera setup in nuScenes. Each missing-view setting involves **multiple combinations** (e.g., 6 for 1 missing view, 15 for 2 missing views, 20 for 3, 15 for 4, and 6 for 5), and SafeMap is trained to generalize across all these cases, maintaining high performance with minimal degradation—credited to its ability to infer missing cues via G-PVR and D-BEVC.
>
> - **Real-World Sensor Corruptions:**
>   - We adopt MapBench [1], which includes eight realistic corruptions across:
>     1. Environmental conditions: `Brightness`, `Low-Light`, `Fog`, and `Snow`;
>     2. Sensor artifacts: `Motion Blur` and `Color Quantization`;
>     3. Sensor failures: `Camera Crash` and `Frame Loss`.
>   - As shown in **Table 9 of the main paper**, SafeMap outperforms MapTR by `+1.9%` mRR / `+9.4%` mCE and HIMap by `+6.2%` mRR / `+16.8%` mCE, demonstrating robustness beyond simple view dropouts and toward real-world deployment scenarios.
>
> ---
> > ***`Q2`: "In Eq (2), why do pixel `px` and pixel `py` follow different noise distributions?"***
>
> A: Thank you for this insightful question. Our design choice is based on empirical observation and domain priors in autonomous driving:
> - The **horizontal coordinate** `px` follows a **Gaussian distribution** centered on the middle of the panoramic perspective view. This reflects the fact that **adjacent views to the missing one contain the most informative features for reconstruction**. The Gaussian distribution allows us to focus attention on these nearby views while still softly exploring farther regions.
> - The **vertical coordinate** `py` follows a **uniform distribution**. This is because **vertical information** (e.g., lane boundaries, poles, road textures) is uniformly important across different heights in the image, especially in BEV-centric tasks like HD map construction.
>
> This **hybrid sampling strategy** balances spatial focus and coverage—allowing G-PVR to direct more attention toward informative horizontal regions while still preserving vertical completeness. Appendix C provides a **sensitivity analysis** showing that this setup offers superior performance over uniform sampling in both dimensions.
>
> ---
> > ***`Q3`: "The backbone builds on MapTR. What is the motivation of the architecture?"***
>
> A: We appreciate this concern. While SafeMap builds upon MapTR for its base encoder-decoder structure, our contributions lie in robust map construction under incomplete observations, which MapTR and existing works are not designed to address. Specifically, our architectural novelty includes:
> - The **G-PVR module**, which performs dynamic reference point sampling using Gaussian priors and deformable attention to reconstruct missing views;
> - The **D-BEVC module**, which distills global spatial priors from complete panoramic views and corrects BEV representations derived from incomplete inputs.
>
> These modules are **model-agnostic** and can be **plugged into other backbones** (e.g., StreamMap), offering a general solution for improving robustness. As shown in the experiments below, SafeMap consistently outperforms both **MapTR** and **StreamMap** variants when camera views are missing:
>
> - *Table. Performance on individual missing views (nuScenes, MapTR backbone):*
> |**View Missing**|**Front**|**Front-L**|**Front-R**|**Back**| **Back-L**|**Back-R**
> | :-|:-:|:-:|:-:|:-:|:-:|:-:|
> |MapTRv2|40.3|53.7|54.2|41.2|56.3|57.4|50.5|
> |**SafeMap**|**53.6**|**59.7**|**59.1**|**52.3**|**60.1**|**60.8**
> | **$\Delta$** |**+12.3**|**+6.0**|**+4.9**|**+11.1**|**+3.8**|**+3.4**
>
> - *Table. Extension to StreamMap (temporal BEV backbone):*
> |**View Missing** | **Front** | **Front-L**  |**Front-R**|**Back**| **Back-L** | **Back-R**|
> | :-|:-:|:-:|:-:|:-:|:-:|:-:|
> |StreamMap (w/o temp)|13.2|17.4|18.0|14.1|16.3|17.1
> |StreamMap (w/ temp)|14.3|18.2|19.3|15.3|17.5|18.5
> |**SafeMap (w/ temp)**|**18.7**|**21.5**|**21.2**|**19.3**|**19.8**|**22.3**
>
> These improvements verify that SafeMap is not merely an application of MapTR, but rather a **general framework** that enhances robustness and reliability under **real-world** constraints, a critical capability not addressed by prior methods.
>
> ---
> **Reference:**
>
> [1] Hao et al. Is Your HD Map Constructor Reliable under Sensor Corruptions? NeurIPS, 2024.

---

### Official Review · Reviewer_smrB · 2025-03-21

**Overall Recommendation:** 3

**Summary:**

The paper introduces SafeMap, a novel framework designed to enhance the robustness of high-definition (HD) map construction for autonomous driving, particularly in scenarios where camera views are incomplete or missing. The key contributions of SafeMap are two innovative modules: the Gaussian-based Perspective View Reconstruction (G-PVR) and the Distillation-based Bird's-Eye-View Correction (D-BEVC). The G-PVR module leverages Gaussian-based sparse sampling to dynamically prioritize the most informative regions from available camera views, while the D-BEVC module uses complete panoramic BEV features to correct BEV representations derived from incomplete observations. The authors demonstrate that SafeMap significantly outperforms existing methods in both complete and incomplete observation scenarios, offering a plug-and-play solution that integrates seamlessly into existing systems. The paper also provides extensive experimental validation on the nuScenes and Argoverse2 datasets, showcasing the framework's superior performance and robustness.

**Claims And Evidence:**

The claims in the submission are largely supported by clear and convincing evidence. The authors provide extensive quantitative results on benchmark datasets (such as nuScenes and Argoverse2) that demonstrate significant improvements in metrics like mAP under both complete and incomplete observation scenarios. In addition, thorough ablation studies isolate the contributions of the Gaussian-based Perspective View Reconstruction (G-PVR) and Distillation-based BEV Correction (D-BEVC) modules, reinforcing their effectiveness in reconstructing missing views and correcting BEV features. Results comparing with state-of-the-art methods, as well as experiments on robustness against camera sensor corruptions (e.g., using MapBench), further substantiate their claims.

**Essential References Not Discussed:**

No

**Experimental Designs Or Analyses:**

Please refer to Claims And Evidence part.

**Methods And Evaluation Criteria:**

The proposed methods and evaluation criteria are well-suited to address the challenges of robust HD map construction for autonomous driving. The approach integrates a Gaussian-based Perspective View Reconstruction (G-PVR) module and a Distillation-based BEV Correction (D-BEVC) module to effectively restore missing view information and correct BEV features derived from incomplete observations, a critical need given the real-world occurrence of sensor failures and occlusions. Moreover, the evaluation criteria—including average precision under various Chamfer distance thresholds, along with experimental validation on established benchmark datasets like nuScenes and Argoverse2—provide a comprehensive measure of both spatial accuracy and overall system robustness. Together, these methods and evaluation metrics offer a balanced and rigorous framework that addresses the core challenges of ensuring accurate and reliable map construction in complex autonomous driving scenarios.

**Other Comments Or Suggestions:**

NA

**Other Strengths And Weaknesses:**

## Paper Strengths
The introduction of the G-PVR and D-BEVC modules is a significant technical advancement. The G-PVR module's use of Gaussian-based sparse sampling to reconstruct missing views is particularly innovative, as it dynamically focuses on the most informative regions across available views. The D-BEVC module's use of complete panoramic BEV features to correct incomplete observations is also a novel approach that enhances the robustness of HD map construction.

The paper addresses a critical challenge in autonomous driving—handling incomplete or missing camera views—which is a common issue in real-world applications. The proposed framework demonstrates strong performance even when crucial camera views are missing, making it highly applicable to practical autonomous driving systems.

The authors provide extensive experimental validation on two widely used datasets (nuScenes and Argoverse2). The results show significant improvements over state-of-the-art methods, particularly in scenarios with missing camera views. The ablation studies further validate the effectiveness of each module, providing strong evidence for the framework's robustness.

SafeMap is designed to be easily integrated into existing systems, which is a practical advantage for real-world deployment. The framework does not require significant changes to existing pipelines, making it a highly attractive solution for enhancing the robustness of HD map construction.

## Major Weaknesses
While the paper focuses on camera-based HD map construction, it does not address the integration of other sensor modalities (e.g., LiDAR, radar) that are commonly used in autonomous driving systems. This limitation could impact the framework's effectiveness in real-world scenarios where multi-sensor fusion is critical for robust perception.

Although the framework is validated on standard datasets, there is no discussion or validation of its performance in real-world deployment scenarios. Real-world conditions, such as dynamic environments, varying lighting conditions, and sensor noise, could pose additional challenges that are not captured in the current experiments.

**Questions For Authors:**

1. Could the authors discuss the potential for extending SafeMap to incorporate multi-sensor fusion, particularly LiDAR and radar? How might this integration affect the framework's robustness and performance in real-world scenarios?
2. The G-PVR module uses Gaussian-based sparse sampling. Could the authors provide more detailed explanations or visualizations of this process?

**Relation To Broader Scientific Literature:**

The paper builds on existing multi-view camera-based HD map construction methods, which have gained attention due to advancements in Bird's-Eye-View (BEV) perception (e.g., MapTR, HIMap). However, it addresses a critical limitation of these methods—their reliance on complete multi-view images—by introducing the G-PVR and D-BEVC modules. The G-PVR module leverages Gaussian-based sparse sampling, a novel approach that dynamically prioritizes informative regions from available views, which is a significant departure from traditional methods that rely on uniform sampling or cropping adjacent views. The D-BEVC module, which uses complete panoramic BEV features to correct incomplete observations, is also a novel contribution that enhances robustness. These innovations are particularly relevant given recent works like MetaBEV and UniBEV, which have started to tackle sensor failures but still rely on complete multi-view images. By addressing the challenge of incomplete observations, SafeMap fills a gap in the literature and provides a robust solution that aligns with the broader trend of enhancing the reliability of autonomous driving systems.

**Theoretical Claims:**

This is an application paper.

---

> ### Author Rebuttal · Authors · 2025-03-28
>
> We sincerely thank the reviewer for the thoughtful and constructive feedback. We greatly appreciate your recognition of our
> *"technical novelty"*, *"robustness in incomplete scenarios"*, *"experimental design"*, and *"plug-and-play applicability"*. Below, we address your main questions and concerns in detail.
>
> ---
> > ***`Q1`: "Extending SafeMap to incorporate multi-sensor fusion."***
>
> A: Thank you for the insightful suggestion. To explore the effectiveness of SafeMap in multi-sensor fusion settings, we extended our framework to **MapTR-Fusion**, a recent **camera-LiDAR fusion** model, and evaluated it under **13 real-world multi-sensor corruptions** introduced in MapBench [1].
>
> The results, presented in the following table, indicate that our SafeMap method not only enhances performance in scenarios involving camera view loss but also significantly boosts the efficacy of **multi-sensor fusion** methods.
>
> |**Method** |Cam. Miss | Cam. Crash  |Frame Lost | LiDAR Miss| Incom Echo | Cross Talk|Cross Sensor|Crash +Echo |Crash +Talk|Crash +Sensor|Frame +Echo| Frame +Talk|Frame +Sensor|Avg.|
> | :-|:-:|:-:|:-:|:-:|:-:|:-:|:-:|:-:|:-:|:-:|:-:|:-:|:-:|:-:|
> |MapTR|22.5|39.1|36.3|20.4|55.1|41.5|39.6|38.6|22.7|20.8|35.8|20.8|19.1|31.7
> |**SafeMap**|**27.6**|**42.1**|**39.8**|**23.6**|**58.3**|**46.2**|**42.1**|**41.3**|**26.7**|**22.7**|**37.2**|**25.3**|**22.7**|**35.0**
> | **$\Delta$** |**+5.1**|**+3.0**|**+3.5**|**+3.2**|**+3.2**|**+4.7**|**+2.5**|**+2.7**|**+4.0**|**+1.9**|**+1.4**|**+4.5**|**+3.6**|**+3.3**
>
> These results further strengthen the practical value of our approach in real-world autonomous driving systems. As suggested, we have added this discussion and table to the revised version.
>
> ---
> > ***`Q2`: "Could the authors discuss real-world deployment scenarios?"***
>
> A: We appreciate your comment on real-world applicability. While large-scale real-world deployments are resource-intensive, we adopted MapBench [1] to systematically simulate **real-world conditions**. This benchmark includes eight **realistic camera sensor corruptions**, categorized into:
> 1. Exterior environmental conditions (`Brightness`, `Low-Light`, `Fog`, and `Snow`).
> 2. Interior sensor artifacts (`Motion Blur` and `Color Quantization`).
> 3. Complete sensor failures (`Camera Crash` and `Frame Loss`).
>
>   As presented in **Table 9 of the main paper**, SafeMap consistently outperforms baselines under these settings:
> - `+1.9%` in mRR and `+9.4%` in mCE over **MapTR**.
> - `+6.2%` in mRR and `+16.8%` in mCE over **HIMap**.
>
> These results clearly demonstrate our robustness under practical corruptions and affirm its suitability for real-world deployment. As suggested, we have included a more explicit discussion in the revised manuscript.
>
> ---
> > ***`Q3`: "More detailed explanations of Gaussian-based sparse sampling."***
>
> A: Thanks for your comments. Our **G-PVR module** employs Gaussian-based sparse sampling to guide reference point selection in the reconstruction of missing camera views. The intuition is to **prioritize sampling locations near adjacent views**, which typically contain the most informative cues for reconstructing the missing view.
>
> Concretely, the horizontal coordinate of reference points is sampled as:
>
> $p_x ∼ 𝓝(μ, σ²),$
>
> and the vertical coordinate is sampled as:
>
> $p_y ∼ 𝕌(0, H),$
>
> where:
> - $μ$ is set to the center of the panoramic perspective view $(μ = (N_a × W) / 2)$.
> - $σ²$ is a hyperparameter controlling the horizontal spread (tuned via ablation in Appendix C).
> - $H$ is the vertical resolution of the input image.
>
> These coordinates define a 2D sampling grid, where reference points are more densely sampled around the central horizontal region (typically where adjacent views reside), and uniformly along the vertical axis.
>
> This sampling pattern is utilized in the Deformable Attention module [2]. This mechanism allows the model to dynamically focus on the most informative regions, concentrating attention near adjacent views while still capturing a broader context. As shown in **Table 5 and Figure 4 in the main paper**, this targeted sampling strategy leads to improved reconstruction quality compared to uniform or local sampling strategies.
>
> ---
> **Reference:**
>
> [1] Hao et al. Is Your HD Map Constructor Reliable under Sensor Corruptions? NeurIPS, 2024.
>
> [2] Zhu et al. Deformable DETR: Deformable Transformers for End-to-End Object Detection, ICLR 2021.

---

> > ### Comment · Reviewer_smrB · 2025-04-04
> >
> > Thanks for the authors' rebuttal. My concerns have been addressed. I will keep my score and support acceptance.

---

> > > ### Author Response · Authors · 2025-04-05
> > >
> > > Dear Reviewer `smrB`,
> > >
> > > Thank you again for your recognition and support!
> > >
> > > The Authors of Submission 8037

---

### Decision · Program_Chairs · 2025-05-01

**Decision:**

Accept (poster)

**Comment:**

The ratings for this paper were all weak accept. All reviewers acknowledged the paper’s novelty and effectiveness in HD map reconstruction under incomplete camera observations and its practical relevance for autonomous driving. The major concerns are about the insufficient experiments on more complex scenarios, such as real-world data (all reviewers) and incorporate with different sensors (Reviewer smrB). Additionally, reviewers raised some technical questions regarding specific details, to which the authors provided satisfactory responses in their rebuttal with further clarifications. All reviewers subsequently agreed to maintain their weak accept recommendations. Based on this assessment, the AC recommends acceptance.